# A Comprehensive QSAR Study on Antileishmanial and Antitrypanosomal Cinnamate Ester Analogues

**DOI:** 10.3390/molecules24234358

**Published:** 2019-11-28

**Authors:** Freddy A. Bernal, Thomas J. Schmidt

**Affiliations:** Institut für Pharmazeutische Biologie und Phytochemie (IPBP), Westfälische Wilhelms Universität-Münster, PharmaCampus-Corrensstraße 48, D-48149 Münster, Germany; f.bernal@uni-muenster.de

**Keywords:** cinnamate ester analogues, QSAR, leishmaniasis, human african trypanosomasis, validation, MLR, OPLS.

## Abstract

Parasitic infections like leishmaniasis and trypanosomiasis remain as a worldwide concern to public health. Improvement of the currently available drug discovery pipelines for those diseases is therefore mandatory. We have recently reported on the antileishmanial and antitrypanosomal activity of a set of cinnamate esters where we identified several compounds with interesting activity against *L. donovani* and *T. brucei rhodesiense*. For a better understanding of such compounds’ anti-infective activity, analyses of the underlying structure-activity relationships, especially from a quantitative point of view, would be a prerequisite for rational further development of such compounds. Thus, quantitative structure-activity relationships (QSAR) modeling for the mentioned set of compounds and their antileishmanial and antitrypanosomal activity was performed using a genetic algorithm as main variable selection tool and multiple linear regression as statistical analysis. Changes in the composition of the training/test sets were evaluated (two randomly selected and one by Kennard-Stone algorithm). The effect of the size of the models (number of descriptors) was also investigated. The quality of all resulting models was assessed by a variety of validation parameters. The models were ranked by newly introduced scoring functions accounting for the fulfillment of each of the validation criteria evaluated. The test sets were effectively within the applicability domain of the best models, which demonstrated high robustness. Detailed analysis of the molecular descriptors involved in those models revealed strong dependence of activity on the number and type of polar atoms, which affect the hydrophobic/hydrophilic properties causing a prominent influence on the investigated biological activities.

## 1. Introduction

Leishmaniasis and human African trypanosomiasis (HAT) are so-called neglected tropical diseases (NTDs), which threaten the life of millions of people around the world [1]. Their burden as well as comorbidity, especially with HIV/AIDS, are well recognized [2,3]. These two vector-borne diseases are caused by *Leishmania* spp. and *Trypanosoma brucei* subspp., respectively, trypanosomatid parasites belonging to the Kinetoplastida order [4,5]. Both leishmaniasis and HAT have received more attention in recent years, yet their currently available treatments present many issues such as, lack of efficacy, increasing resistance, and high toxicity [6,7], making it necessary to strengthen the research efforts in this field. Various compound types have demonstrated potential *in vitro* against these parasites [8]. Nonetheless, the number of new compounds entering clinical trials is regrettably low [7] and the search for antitrypanosomatid agents therefore remains an important issue [6,9,10].

Studies on quantitative structure-activity relationships (QSAR) constitute an important computational tool in drug discovery, which can help with extracting useful information from big data matrices and thus, guiding in a more rational and structured way the drug design process. Some recent applications of QSAR in virtual screening for anti-infective drugs have been mentioned [11]. More specifically, QSAR approaches against NTDs, including leishmaniasis and HAT, have also been reported and compiled [12]. On the other hand, the potential of natural derivatives and synthetic analogues of cinnamic acid for combating leishmaniasis and trypanosomiasis have been widely described in the few past years [13,14,15,16,17,18,19,20,21,22,23,24,25]. Therefore, and as part of our continuous efforts to fight NTDs, we present a comprehensive QSAR study on a set of synthetic esters of the natural product cinnamic acid, which we recently described as potent and selective agents against *L. donovani* and *T. brucei rhodesiense* [25]. Even though those compounds may not strictly fulfill the stringent criteria for “hits” proposed by Katsuno et al. [26], a deeper understanding of the underlying structure-activity relationships (SARs) from the quantitative point of view together with the predictive ability of robust QSAR models may result in an actual hit compound. Thus, the present research aimed at the building, selection, validation, and interpretation of QSAR models independently predicting antileishmanial and antitrypanosomal activity of the cinnamate ester analogues. A set of newly introduced scoring functions accounting for fourteen validation parameters and different validation criteria currently available was used as a key tool to define the best models.

## 2. Results and Discussions

### 2.1. Cinnamate Ester Analogues and Their Molecular Fingerprints

The set of compounds included in the present study is shown in Figure 1. The structures were used to calculate the respective Molecular ACCess System (MACCS) 166-bit fingerprints, which were primarily used to look for general relationships between simple constitutional structural features and antiparasitic potential. Those keys were firstly compared by principal component analysis (PCA) as shown in Figure 2A. Compounds are color-coded in groups resulting from a hierarchical clustering analysis (HCA) on the same data. The fingerprints for the nitro derivatives **20**–**23** were clearly distinguished from all other compounds on the left side of the score plot at low scores on the first principal component (PC1; green). The esters without oxygenation on the aromatic ring (**29**–**33**) formed a second separate group with low scores on PC2 (blue). The fingerprints for the rest of the compounds were too close to be discriminated on the PC1–PC2 score plot. However, the compounds in red and yellow were consistently discriminated from each other by the third principal component (Appendix A) which appeared to be related to the nature of the ester side chain. Thus, methyl esters as well as those compounds bearing branched lateral chains were grouped together (yellow), whereas those with linear chains bigger than methyl were independently clustered (red).

Furthermore, the MACCS fingerprints and activity values against *L. donovani* and *T. brucei rhodesiense* were independently analyzed by partial least squares (PLS) regression (the activity data are found in Appendix A). The good statistical validation of both models ($R^{2}$ = 0.897; $Q^{2}$ = 0.645 and $R^{2}$ = 0.906; and $Q^{2}$ = 0.666, for antileishmanial and antitrypanosomal activity, respectively) clearly proved the dependence of the respective biological activity on structural characteristics encoded by this fingerprint. Use of orthogonal partial least squares (OPLS) instead of simple PLS led to similar models but with even better cross-validation results ($R^{2}$ = 0.897; $Q^{2}$ = 0.782 and $R^{2}$ = 0.935; and $Q^{2}$ = 0.798, for antileishmanial and antitrypanosomal activity, respectively). Figure 2B,C shows the corresponding plots of OPLS-predicted versus observed activities. Comparison of the effect of each independent variable on the activity as well as evaluation of the direct correlation between them can be retrieved from the corresponding S-lines as implemented in SIMCA [27]. Analysis of the S-lines (Figure 2D) revealed that the structural keys contributing most to the variance in antileishmanial activity, and at the same time highly correlating with it, were numbers 140, 144, 150, 147, and 132. Those bits from the MACCS fingerprint would be likely responsible for the activity. Such features are related to the number of oxygen atoms in the molecule and to the existence of linear alkyl chains, which is in perfect agreement with the previously published qualitative SAR results [25], but also led to inferring that the antileishmanial activity is overall affected by changes in polarity. The same analysis for the antitrypanosomal activity-based model (Figure 2C) indicated that the most important keys in that case are 54, 131, 127, and 72 (Figure 2E), which are mainly related to the presence of geminal oxygens (i.e., carboxyl, nitro, etc.) and heteroatoms bonded to hydrogen (i.e., hydroxyl groups) rather than just the presence of oxygen as in case of antileishmanial activity. The antiparasitic activity data of the esters was therefore also investigated for correlations with physicochemical descriptors related to polarity, namely, the calculated octanol/water partition coefficient (cLogP) and the topological polar surface area (TPSA), both computed using the Molecular Operating Environment software (MOE) [28]. Whereas the antitrypanosomal activity showed no correlation with these calculated properties, the antileishmanial activity appeared to be linearly correlated with TPSA (Pearson coefficient, *R* = 0.779), i.e., in this series, higher polarity apparently tends to increase the antileishmanial activity (compound **18** appeared as a potential outlier of a relatively well-defined trend; Figure 2F). This interesting result encouraged us to perform a more detailed QSAR analysis for this dataset.

### 2.2. QSAR Modeling for Antileishmanial Activity

#### 2.2.1. Building of QSAR Models for Antileishmanial Activity

3D models of the thirty-four compounds shown in Figure 1 were prepared as follows. Each structure was independently submitted to conformational search using the low molecular dynamics (LowMD) mode in MOE [28] within an energy window of 5 kcal/mol. The structures of the resulting lowest energy conformers were subsequently refined using the semi-empirical Austin model 1 (AM1) method and used to calculate a set of 435 molecular descriptors available in MOE. Afterwards, a contingency analysis as implemented in MOE was carried out to identify the descriptors with highest utility for QSAR modeling in the current data set. Descriptors with contingency coefficient above 0.6 and Cramer’s uncertainty values as well as correlation coefficients above 0.2 were then selected to be included in the QSAR modeling process (158 descriptors). The set of compounds was independently divided twice to yield two different training and test sets by random selection. Additionally, a third division into training and test set was obtained based on maximum dissimilarity using the Kennard-Stone algorithm [29,30,31,32]. The ratio training/test set was kept constant in all three sets (*N* = 26 for training and *N* = 8 for test set). QSAR model building for each training set was performed by means of multiple linear regression (MLR) using a genetic algorithm (GA) for selection of the best descriptors (the application of genetic algorithms in QSAR is well-established [33,34,35]). Fixed model lengths were always employed during GA/MLR modeling, with three, four, and five descriptors per model. The highest number of descriptors per model was chosen following the early observations of Topliss and Costello [36], and later adopted as a rule of thumb by the Organisation for Economic Co-operation and Development (OECD), in which a minimum ratio of five objects per selected variable is recommended to avoid chance correlations and overfitting [35]. Each resulting family of models was cross-validated (CV) by the leaving-one-out method (LOO). The models were then sorted by their $Q^{2}$ values and the best five models with the highest $Q^{2}$ values within each family were selected for further analysis. This approach led to a total set of 45 different QSAR models describing the antileishmanial activity of the cinnamate ester analogues (15 models for each training set). The increasing $Q^{2}$ from three to five descriptors for models using the training set from Kennard-Stone method led to generate an additional family of models using also six descriptors during GA/MLR, reaching a total of 50 models. Thus, the total number of possible appearances of descriptors in the overall set of 50 QSAR equations was {[(3 × 3) + (3 × 4) + (3 × 5) + (1 × 6)] × 5} = 210. Descriptors ASA^–^ and PEOE_VSA_FPOL occurred most frequently, appearing in 12.9% and 11.4%, respectively, of all cases (Table 1), which corresponds to 54% and 48% appearance in the set of 50 equations, respectively. These two descriptors were followed by chi1v, PEOE_VSA_FHYD, and vsurf_D1. Besides those five descriptors with the highest frequency in the equations, another 62 out of the 158 descriptors were included in some of the considered models. Interestingly, the descriptor TPSA, quite strongly correlated with antileishmanial activity as mentioned above, occurred only in one single equation. Nonetheless, four out of the five most frequent descriptors (Table 1; all except chi1v, which is a topological descriptor related to molecular shape) are in fact, like TPSA, related to the polarity of the molecules. A more detailed analysis of the descriptors involved in the models will be presented below, after model validation and selection of the best models according to their predictive performance.

#### 2.2.2. Validation of QSAR Models for Antileishmanial Activity

In order to assess the models’ statistical validity, a set of validation parameters was used which comprises all methods proposed in literature for this purpose that have come to the authors’ attention. Those parameters are: $R^{2}$, $Q^{2}$, the Golbraikh and Tropsha criteria [37] (considering $R_{0}^{2}$, $R_{0}^{{}^{\prime }2}$, *k*, and *k*’), both criteria introduced by Roy et al. (using $R_{m}^{2}$ [38,39] and using the mean absolute error (MAE) [40]), $Q_{F1}^{2}$ of Shi et al. [41], $Q_{F2}^{2}$ of Schüürmann et al. [42], $Q_{F3}^{2}$ of Consonni et al. [43], and the concordance correlation coefficient (CCC) adapted by Chirico and Gramatica for QSAR purposes [44] (see Appendix A for definitions). Instead of making a direct comparison of all of those parameters for the whole set of models as performed by previous authors [45], some scoring functions accounting for each model’s compliance with the diverse criteria were defined in the present research, as follows (Table 2): *F*1 is the sum of the number of parameters that fulfilled the respective condition and/or threshold to be a good model, giving a general insight about the quality of the models in terms of their statistical validity. The maximum number of parameters to pass (i.e., the maximum of *F*1) is 12. A more precise discriminant result was obtained by the scoring function *F*2 which considers the differences between a model’s values for the specific parameters and their defined thresholds (i.e., the value of *F*2 reflects a model’s quality beyond the established thresholds); and finally, *F*3 introduces the number of descriptors per model as a selection criterion. It is a fact that a smaller number of variables generally helps to avoid overfitting and also improves a model’s interpretability, an important characteristic of QSAR models which is often neglected [46,47]. *F*3 was exclusively calculated for models that offered good quality on their predictions according to the criteria of Roy et al. [40] (i.e., criteria using MAE). Comparable approaches involving most of the statistical validation parameters presented here have been already described in literature, e.g. [48], which were typically based on the use of the QSARINS software developed by Gramatica et al. [49].

To obtain *F*1, the threshold values originally proposed were kept in this work as reported in Table 2. On the other hand, *F*2 included an eighth *P*1 element which compares $R^{2}$ with $Q_{F2}^{2}$ (also known as $R_{pred}^{2}$) and rewards the fact of having higher $R^{2}$ than $Q_{F2}^{2}$ and penalizing the contrary, i.e., it attempts to assure that the fitting of the external set of compounds did not overpass that for the training set. This penalty was introduced since very high $Q_{F2}^{2}$ values exceeding the overall $R^{2}$ of the training set may be related to overfitted models. The first *P*2 element in *F*2 is selected according to the lowest difference between either $R_{0}^{2}$ or $R_{0}^{{}^{\prime }2}$ and $R^{2}$. The apparently most complicated terms in *F*2 are related to the criteria using MAE (*P*3). The first *P*3 element corresponds to the penalization of the models for offering poor predictions according to the original criteria proposed by Roy et al. [40], using the highest value between $MAE-0.15*pIC_{50}\;range$ and $MAE+3\sigma_{MAE}-0.25*pIC_{50}\;range$. The second *P*3 element has a twofold effect, rewarding models characterized as “good” according to the original criteria and penalizing those, which fail it. The resulting score values are shown in Table 3 (Appendix A presents the whole set of statistical validating parameters for each model).

As can be seen from Table 3, only few models resulted in failure to many of the validation criteria (showing therefore low *F*1 values). In most of the cases, models with *F*1 = 12 corresponded to those with the highest *F*2 values, too. There were however some exceptions like models M1-3, M3-3, M5-5, M6-3, M6-1, and M9-2, which exhibited high *F*2 without fulfilling all the validation criteria (i.e., F1 < 12). The highest *F*2 scores were mainly displayed by models coming from the second random selection. Specifically, M6-2 was defined as the best model in terms of the validity of its predictions for both, the training set as well as the test set. The wide difference in *F*2 scores demonstrates the well-known effect of the selection of the training and test sets and the selection of the descriptors used to build the model. Interestingly, when a low number of descriptors was considered as a quality criterion (*F*3), M5-1 was highlighted as the best model, conferring an excellent compliance of all the validation parameters (*F*1 = 12) but at the same time showing the best ratio validity score/number of descriptors. Plots of the predicted versus the experimental antileishmanial activity for the training and test sets of the two mentioned models of highest quality, M6-2 and M5-1, together with M2-1 and M10-1 as best representatives of the models using the first random selection and the Kennard-Stone algorithm, respectively, are presented in Figure 3.

It becomes obvious from Figure 3 that the four selected models performed comparably well. However, the trend line for the test set in M10-1 is more deviated than that for the others, which means M10-1 (Figure 3D) underestimates the antileishmanial activity of external compounds in spite of a very good linear correlation with the actual activity (i.e., high $Q_{F2}^{2}$). On the other hand, M2-1 (Figure 3A) exhibited relatively high both positive and negative deviations in the predicted activity when limiting values are considered. Visual inspection of these plots confirmed the superior performance of models M5-1 and M6-2 (Figure 3B,C, respectively).

#### 2.2.3. Robustness and Applicability Domain (AD) Definition of the Best QSAR Models for Antileishmanial Activity

The noteworthy better performance of models M5-1 and M6-2 qualified them for assessment of their robustness using a 100-run Y-scrambling test [50,51] (Figure 4A,B) and for determination of their applicability domain (AD) by means of the leverage method [51,52] (Figure 4C,D).

The Y-scrambling test (100 runs) displayed comparable results for both models (Figure 4A,B). Overall, the $R_{rand}^{2}$ and $Q_{rand}^{2}$ values (i.e., $R^{2}$ and $Q^{2}$ values after randomization of the Y response) were always much lower than the actual $R^{2}$ and $Q^{2}$ values for the “true” (unscrambled) models, and also lacked statistical significance. This demonstrates that there was no chance correlation between the corresponding molecular descriptors involved in those models and the antileishmanial activity. Moreover, the correlation between the original Y response and the scramble response were mainly below 0.4 (with a few exceptions), reinforcing the robustness of both models. Their similar behavior is in agreement with the previously shown statistical validity and with the fact that both models were obtained with the same training and test sets. Nonetheless, both models appeared differentially influenced by the compounds in the training set as observed from their very different distribution in the corresponding Williams plots (Figure 4C,D). Compound **28** had a strong influence on both models, although still with acceptably low leverage. Model M6-2 additionally was significantly influenced by compound **18**. Compounds **26** and **31** showed relatively high standardized residuals in Model M6-2 (Figure 4D), but these deviations were below 2.5σ and the corresponding predictions could still be considered as acceptable [52]. Most importantly, all the compounds in the test set were effectively within the AD of both models (all the red diamonds appeared below the critical leverage value, *h**), mainly with low standardized residuals.

#### 2.2.4. Interpretation of the Best QSAR Models for Antileishmanial Activity

Table 4 presents the QSAR equations of the two best performing models M5-1 and M6-2. These models included three common descriptors: ASA^–^, a_IC, and vsurf_W3. ASA^–^ refers to the water accessible surface area of all atoms with negative partial charge; a_IC corresponds to the total atom information content, which is related to the number of occurrences of each element in the molecule; vsurf_W3 is one out of eight different descriptors defining the hydrophilic volume. According to the sign of the respective regression coefficients in both, M5-1 and M6-2, increasing ASA^–^ and vsurf_W3 would have a deleterious result for the antileishmanial activity, whereas increasing a_IC could result in increased activity. Model M5-1 also included the descriptor PEOE_VSA_FPOL, which is the fractional polar van der Waals surface area, whereas M6-2 contained PEOE_VSA_FHYD, the fractional hydrophobic van der Waals surface area, instead. These two descriptors actually represent the same information. From the opposite sign of their regression coefficients, it is clear that higher overall polarity increases and higher hydrophobicity decreases antileishmanial potency. This is complemented in the more complex five-descriptor model M6-2 by the descriptor Q_VSA_POL, which is the total polar van der Waals surface area. Its negative regression coefficient points into the same direction. Even though the descriptors are not correlated with each other, PEOE_VSA_FHYD and Q_VSA_POL (Pearson’s correlation coefficient *R* = −0.151), the latter is probably only of modulatory influence and the benefit of including it in the model (and increasing the number of descriptors over M5-1) appears doubtful.

In general, all the descriptors involved in models M5-1 and M6-2 are in fact related, in one way or another, to the molecules’ content of polar atoms and their ability to generate polar contacts (e.g., H-bonds) which increase the antileishmanial activity. This finding is well in line with the simple OPLS model based on molecular fingerprints (see Section 2.1.) but reflects the impact of this property in a much more detailed manner.

### 2.3. QSAR Modeling for Antitrypanosomal Activity

#### 2.3.1. Building of QSAR Models for Antitrypanosomal Activity

An analogous protocol to the one described above, based on the same training and test set divisions, led to 45 different QSAR models for the antitrypanosomal activity of the cinnamate ester analogues. In this case, of 277 descriptors considered in the GA/MLR modelling process, two descriptors appeared with high frequency in the models, namely vsurf_EWmin3 and std_dim2. Other six descriptors were present in a significant number of the models (Table 5). Quite noteworthy, these descriptors were not part of the models for antileishmanial activity, suggesting that both biological properties are not influenced by the same molecular properties.

#### 2.3.2. Validation of QSAR Models for Antitrypanosomal Activity

The statistical validation and scoring of the models for antitrypanosomal activity was accomplished in the same way as above (see Section 2.2.2.). The individual model scores, *F*1–*F*3, are presented in Table 6. The whole set of validation parameters is compiled in Appendix A.

The GA/MLR QSAR modeling for the antitrypanosomal activity resulted in an apparently higher variability within models, including poor quality models as those using three descriptors for the first randomly generated training set (M11-1 to M11-5; Table 6). The reliability of the scoring function *F*2 was clearly demonstrated with the severe penalization of those models (all of them failed most of the validation criteria as indicated by an extremely poor *F*1 score). Notably, not all the models with the highest *F*2 values matched those with the highest *F*3 score. Similar to the findings for antileishmanial activity, the models for antitrypanosomal activity with better scores were obtained with the randomly selected test sets. M13-3 was found as the best model according to *F*2, while M14-2 was defined as the best based on *F*3. Plots of predicted versus experimental activity for these two models together with M12-4 (highest *F*3 within models with test set one), M16-2 (highest *F*2 within models with test set two), M17-2, and M18-1 (highest *F*3 and *F*2 within models with test set three, respectively) are depicted in Figure 5.

The plots in Figure 5 clearly show the relatively high deviations of external activity predictions by model M18-1 (Figure 5F). In contrast, model M13-3 predicted both training and test set activities with the same accuracy (same slope; Figure 5B). Very close to this behavior appeared models M16-2 and M17-2 (Figure 5D,E). M12-4 (Figure 5A) showed similarly constant but positive deviations of the test set predictions, while M14-2 (Figure 5C) displayed variable deviations along the activity range. On the other hand, the predictions from M13-3 (highest *F*2 score) seem to be somewhat more reliable than those from M14-2 (highest *F*3 score).

#### 2.3.3. Robustness and AD Definition of the Best QSAR Models for Antitrypanosomal Activity

The models for antitrypanosomal activity with highest *F*2 and *F*3 scores, M13-3 and M14-2, respectively, were further investigated for robustness and AD as described above. Figure 6 shows the corresponding results.

A straightforward conclusion from Figure 6A,B is that both, M13-3 and M14-2, are statistically robust models with low correlation between the scrambled and the actual response as well as low statistical significance of the $R_{rand}^{2}$ and $Q_{rand}^{2}$ within 100 runs. Furthermore, all the compounds in the test sets were effectively within the AD of both models (Figure 6C,D). Model M13-3 was strongly influenced by compound **34**, whereas M14-2 was influenced, but to lesser extent, by compounds **28**, **11** and **33**. Predictions for compounds **10** and **18** in M13-3 displayed the highest deviations with standardized residuals, but only slightly above 2σ. The prediction of activity for compound **34** by M14-2 was similarly high but also acceptable (standardized residual below 2.5σ) [52].

#### 2.3.4. Interpretation of the Best QSAR Models for Antitrypanosomal Activity

The QSAR equations of the two models with best performance according to their *F*2 or *F*3 scores are reported in Table 7. It is evident that the two models differ from each other more significantly than those for antileishmanial activity. However, descriptor vsurf_EWmin3, the most frequent variable in the whole set of models, occurs in both of them. This descriptor encodes the interaction energy at one of three lowest local interaction minima of the molecules with a water molecule and is thus a measure for their hydrophilicity or polarity [53]. The negative regression coefficient of this variable in both equations indicates that negative values of the descriptor (i.e., high polarity) will positively influence activity. The lesser complex model, M14-2, besides this, contains descriptor lip_don which represents the number of hydrogen bond donors, i.e., OH and NH groups. Its positive coefficient points towards an enhancing effect of such groups on activity. Besides this, descriptor chi1_C, the 1^st^ order carbon connectivity index, is a topological descriptor representing a measure of molecular complexity in terms of branching within the molecular graph [54,55]. Even though its physical meaning appears somewhat cryptic, its positive coefficient seems to indicate that higher complexity (i.e., more and bigger substituents on the common cinnamate scaffold) increase activity. Model M13-3 instead included the descriptor ASA^–^ with a negative coefficient as the models for antileishmanial activity did. Two further descriptors in M13-3 were Q_RPC^–^ and Q_VSA_FPPOS, which describe the relative negative partial charge (most negative partial charge divided by the sum of negative partial charges) and the fractional positive polar van der Waals surface area (sum of surface areas for atoms with partial charges >0.2 e divided by the total surface area), respectively. Thus, both of these descriptors are representations of molecular polarity. The former, related to electron rich areas of the molecules, has a negative regression coefficient indicating that too much local negative charge leads to decreased activity. The latter with a positive coefficient indicates that the opposite is true for positive partial charge; since H-bond donor protons usually have the highest positive partial charge with values >0.2 e in the molecules under study, this agrees with the contribution of lip_don to the aforementioned model. Thus, polarity—although represented in a somewhat more detailed way in the models—also plays the most prominent role in case of the antitrypanosomal activity of the compounds under study. In this case, the potency to engage in hydrogen bonds appears to be of more explicit importance than overall polarity. This is in agreement with the lack of correlation of the antitrypanosomal activity with the global polarity descriptor TPSA mentioned in Section 2.1

## 3. Materials and Methods 

### 3.1. Data Preparation

Compounds **1**–**34** were recently prepared, characterized, and tested for their activity against *Leishmania donovani* and *Trypanosoma brucei rhodesiense* within our working group [25]. Three-dimensional molecular representations of each compound were created using MOE, Montreal, QC, Canada (version 2018.01) [28] and subsequently optimized by energy minimization using the AMBER10: EHT force field, which is adequately parameterized for small molecules and provided better outcome than typical force fields like MMFF94x (the latter caused torsion of the double bond of the cinnamate, resulting in twisted structures lacking co-planarity of the π orbitals of the conjugated phenyl-acrylate system). The obtained structures were submitted to conformational analysis through the LowMD mode of MOE in an energy window of 5 kcal/mol. The structure of the lowest energy conformer in each case was finally optimized using the semi-empirical AM1 method (MOPAC module in MOE). The geometries thus obtained were employed for the calculation of the 166-bit MACCS fingerprints and the whole set of molecular descriptors offered by MOE. Afterwards, selection of the most suitable molecular descriptors for QSAR purposes was carried out by contingency analysis as implemented in MOE. Descriptors with contingency coefficient above 0.6 and Cramer’s, uncertainty, and correlation coefficients above 0.2 were selected for QSAR (see Appendix A). The biological activity values of the compounds were expressed as the negative decadic logarithms of the IC_50_ values expressed in molarity (pIC_50_).

### 3.2. QSAR Modeling: First Approach Using Molecular Fingerprints

The set of molecular fingerprints (MACCS 166-bits), were initially compared using principal component analysis and hierarchical clustering analysis in SIMCA, Umeå, Sweden (version 14.1) [27]. Afterwards, partial least squares regression and orthogonal partial least squares regression were independently applied on the same dataset using the corresponding biological activity as Y response. In case of OPLS, the S-line plots offered by SIMCA were also analyzed and included herein.

### 3.3. QSAR Modeling by GA/MLR

The datasets obtained as described in Section 3.1. were divided into training and test sets as follows:

*Random selection:* The compounds were sorted in the order of descending activity and eight different bins were defined. From each bin a compound was randomly selected and assigned to the test set (*N* = 8). The process was independently repeated to obtain a second randomly selected test set. 

*Rational selection*: This was performed using an in-house MATLAB script in MATLAB R2018b, Natick, MA, USA [56], based on the descriptor space using maximum dissimilarity following the Kennard-Stone algorithm [31] as indicated by Martin et al. [29]. To do so, the descriptors were normalized and used to define similarity among compounds by measuring Euclidean distances,
(1)$$distance=\sqrt{\overset{w}{\underset{k=1}{\sum }}{\left(X_{ik}^{n}-X_{jk}^{n}\right)}^{2}}$$
where *w* represents the number of descriptors and $X_{ik}^{n}$ and $X_{jk}^{n}$ are the normalized values for the descriptor *k* of compounds *i* and *j*. The normalization was accomplished by subtraction of the mean descriptor value, $\bar{X_{k}}$, from the descriptor for the corresponding compound, $X_{ik}$, and subsequent division by the standard deviation of the descriptor, $\sigma_{k}$:(2)$$X_{ik}^{n}=\frac{X_{ik}-\bar{X_{k}}}{\sigma_{k}}$$

The pair of compounds with the highest distance were then selected for the training set. The next step consisted of selection of the compounds with maximum dissimilarity from each of the two previously selected compounds, in order to be part of the training set. This process was repeated until the number of compounds in the training set had the same size as for the random selection (*N* = 26). The remaining compounds (*N* = 8) were assigned as test set (all the training and test sets are disclosed in Appendix A).

The three different training sets defined for each biological activity were employed to build QSAR models using the genetic algorithm-driven variable selection and multiple linear regression analysis. The GA algorithm was obtained from the CCG/MOE SVL exchange website (script GA.svl) [57]. The number of variables for the models was fixed to values of three, four, five, and six, i.e., the algorithm was independently applied for each training set to each of those model sizes. In each GA/MLR run, a set of 100 models was generated which were ranked by their corresponding $Q^{2}$ values from the leave-one-out cross-validation. Each GA run had a maximum of 1000 evolution cycles as termination criteria. The five best models of each GA run according to their $Q^{2}$ values were selected for further validation. Models with six descriptors were exclusively included when they proved to afford higher $Q^{2}$ values than the five-membered congeners.

### 3.4. Complete Validation of the QSAR Models

The selected models from the GA/MLR analysis were submitted to validation in terms of the quality of their predictions by means of the calculation of $R^{2}$, $Q^{2}$, $R_{0}^{2}$, $R_{0}^{{}^{\prime }2}$, *k*, and *k*’ $R_{m}^{2}$, mean absolute error, $Q_{F1}^{2}$, $Q_{F2}^{2}$, $Q_{F3}^{2}$, and the concordance correlation coefficient. Most of the corresponding definitions are found in literature [58] and described in the Supporting Information (Appendix A). The calculation of all these parameters was simultaneously accomplished using an in-house MATLAB script. All the models were then scored by applying the scoring functions *F*1, *F*2, and *F*3 in a Microsoft Excel, Redmond, WA, USA, spreadsheet. The best two models (one according to *F*2 and one according to *F*3) were finally evaluated for robustness by a Y-randomization test using an in-house MATLAB algorithm. A total of 100 scrambled runs were carried out and the calculated $R_{rand}^{2}$ and $Q_{rand}^{2}$ values were plotted against the correlation coefficient between the scrambled and the actual activity data, as an adaptation of the response permutation plot of SIMCA [27] introduced by Eriksson et al. [59]. The best models based on the scoring were also assessed for their AD by the leverage approach [51,52] calculating the leverage value for each compound, *h*:(3)$$h_{i}=x_{i}^{T}{\left(X^{T}X\right)}^{-1}x_{i}$$
where $x_{i}$ and $x_{i}^{T}$ are the descriptor vector of the query compound and its transposed, respectively, and X and $X^{T}$ are the complete descriptor matrix of the model (*N* compound × *w* descriptors) and its transposed, respectively.

The calculation was done using an in-house MATLAB algorithm. The results were displayed as the corresponding Williams plots showing the critical leverage, *h** (*h** = 3*w*/*N*, where *w* is the number of descriptors in the model plus one and *N* is the number of compounds) [51,52].

## 4. Conclusions

The noteworthy antileishmanial and antitrypanosomal activity of various cinnamate ester analogues among compounds **1**–**34**, in addition to the outstanding selectivity of some of those compounds against the parasites under study in comparison with mammalian cells [25], encouraged the detailed QSAR study presented here. A new set of scoring functions was defined to help selecting the best models within large model populations obtained by GA/MLR with very similar statistical validation parameters. This constituted a first approach to attempt a comprehensive exploitation of the multitude of validation parameters available in literature. This approach can be easily extrapolated to any QSAR modeling situation and represents a more rational means for judging the predictive quality and reliability of QSAR models than highlighting some of them by arbitrarily using a single or a few statistic parameter(s) as often practiced in QSAR studies. In this case, it aided the comprehensive analysis of series of QSAR models obtained by GA/MLR with extraordinary quality of predictions of the antileishmanial and antitrypanosomal activity of the studied compounds. In case of the antileishmanial activity, rather homogeneous model families with identical or closely related descriptors were found irrespective of training/test set division, model size, etc. These descriptors represent key molecular features important for the antileishmanial activity of the cinnamate ester analogues. In case of antitrypanosomal activity the model families were much more heterogeneous which makes their interpretation more demanding. However, also in this case, the very good performance and compliance with all of the validation parameters evaluated was demonstrated. The best QSAR models obtained in this study thus constitute a useful predictive tool to aid the subsequent development of new antiparasitic leads and drugs, based on this type of compounds with a rather simple natural scaffold. Studies in this direction are in progress.

## Figures and Tables

**Figure 1 molecules-24-04358-f001:**
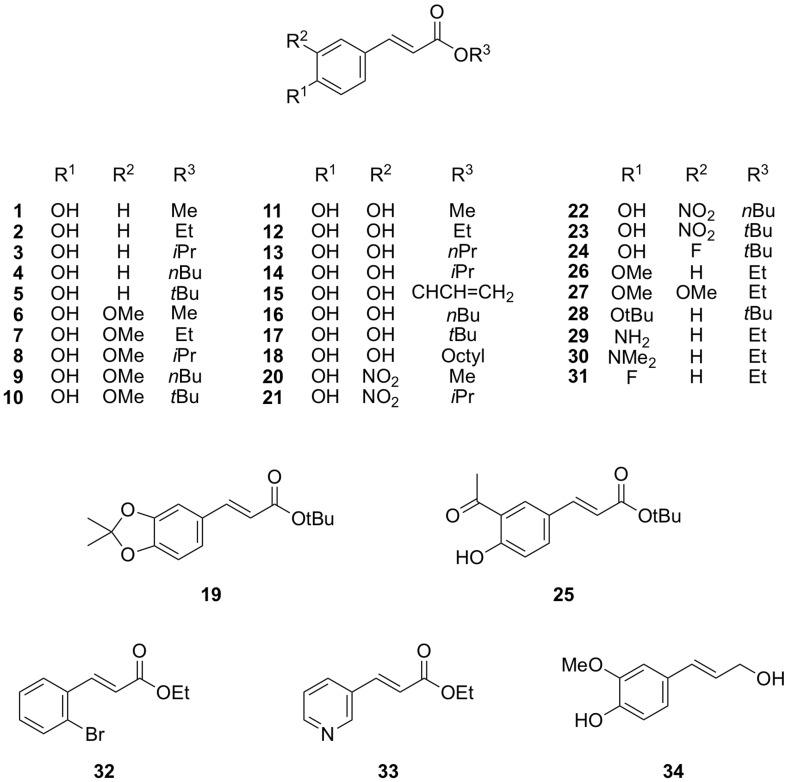
Chemical structures of the analyzed antiparasitic compounds. The preparation and biological evaluation of these compounds was previously reported [25].

**Figure 2 molecules-24-04358-f002:**
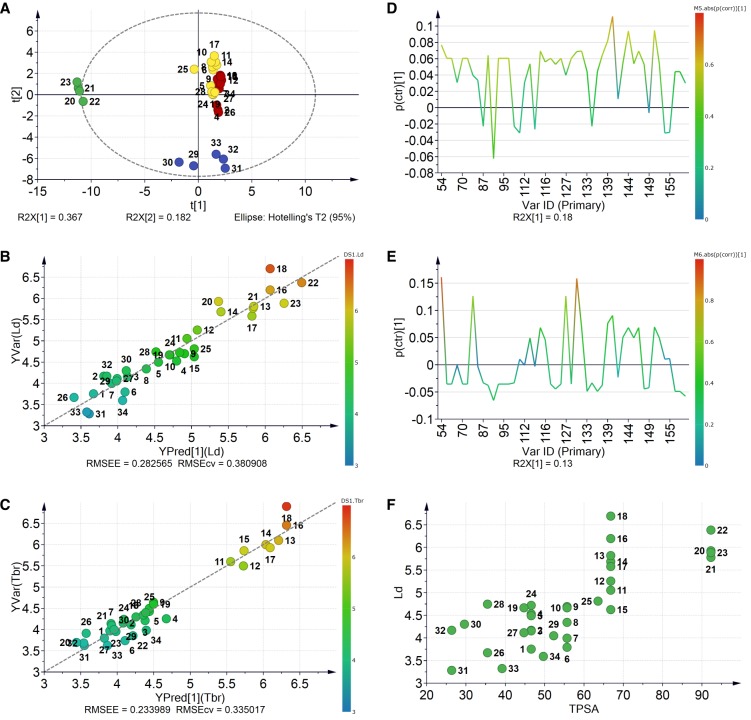
Multivariate statistical analysis of MACCS fingerprints by (**A**) principal component analysis (PCA); (**B**) orthogonal partial least squares (OPLS)-predicted versus experimental antileishmanial and (**C**) antitrypanosomal activity; (**D**) and (**E**) corresponding S-line plots for antileishmanial and antitrypanosomal activity, respectively; and (**F**) variation in antileishmanial activity with topological polar surface area (TPSA). Dashed lines in (**B**) and (**C**) represent linear correlation. (**A**) is color-coded by groups of compounds obtained by hierarchical clustering analysis (HCA); (**B**) and (**C**) are color-coded by the respective activity (pIC_50_ values; numerical values in Appendix A); and (**D**) and (**E**) are color-coded by correlation of each variable with activity.

**Figure 3 molecules-24-04358-f003:**
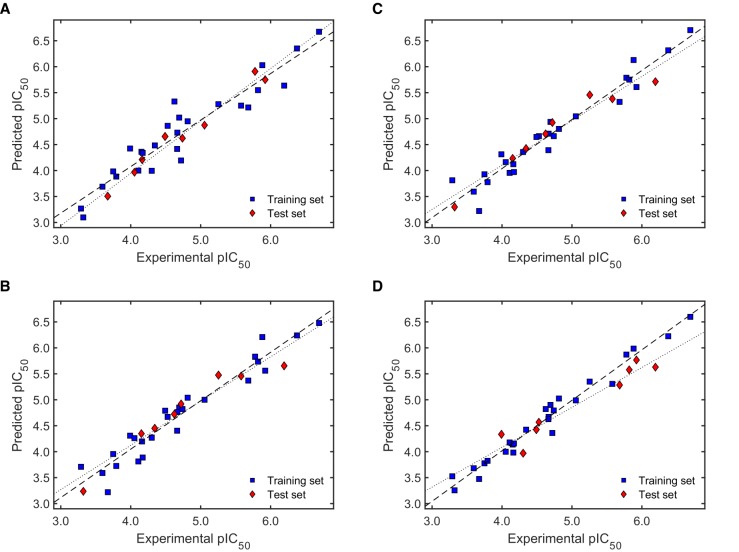
Predicted versus experimental antileishmanial activity plots for models (**A**) M2-1, (**B**) M5-1, (**C**) M6-2, and (**D**) M10-1. Dashed and dotted lines show the linear correlation for the training and test set, respectively. Numerical values for the experimental activity are shown in Appendix A.

**Figure 4 molecules-24-04358-f004:**
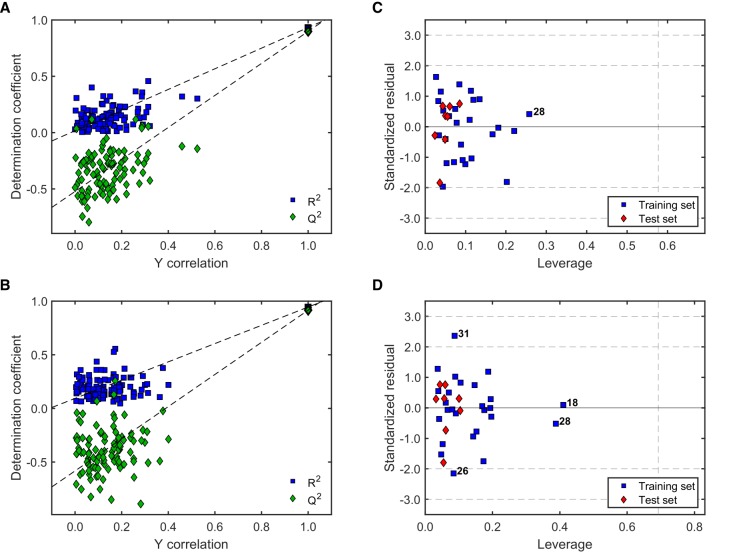
Evaluation of robustness and applicability domain (AD) of the best models for antileishmanial activity. (**A**) Y-scrambling for M5-1; (**B**) Y-scrambling for M6-2; (**C**) Williams plot for M5-1; and (**D**) Williams plot for M6-2. In A and B, $R^{2}$ and $Q^{2}$ values for the “true” (unscrambled) model appear in the upper right corner with thicker marker edge. Horizontal dashed lines in (**C)** and (**D)** indicate 2σ and 3σ. The vertical dashed lines represent *h** [50,51,52].

**Figure 5 molecules-24-04358-f005:**
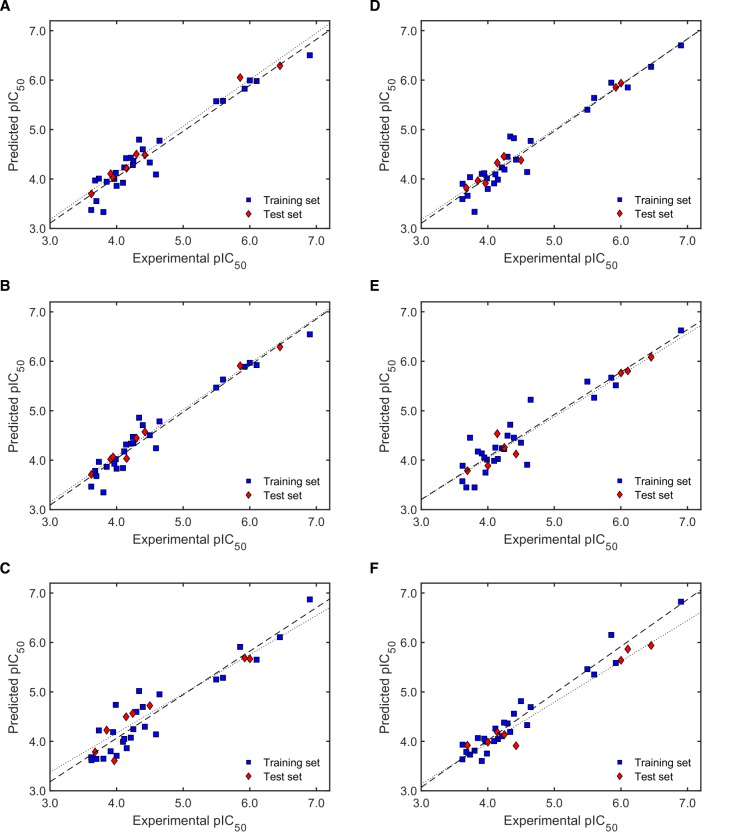
Predicted versus experimental antitrypanosomal activity plots for models (**A**) M12-4, (**B**) M13-3, (**C**) M14-2, (**D**) M16-2, (**E**) M17-2, and (**F**) M18-1. Dashed and dotted lines show the linear correlation for the training and test set, respectively. Numerical values for the experimental activity are shown in Appendix A.

**Figure 6 molecules-24-04358-f006:**
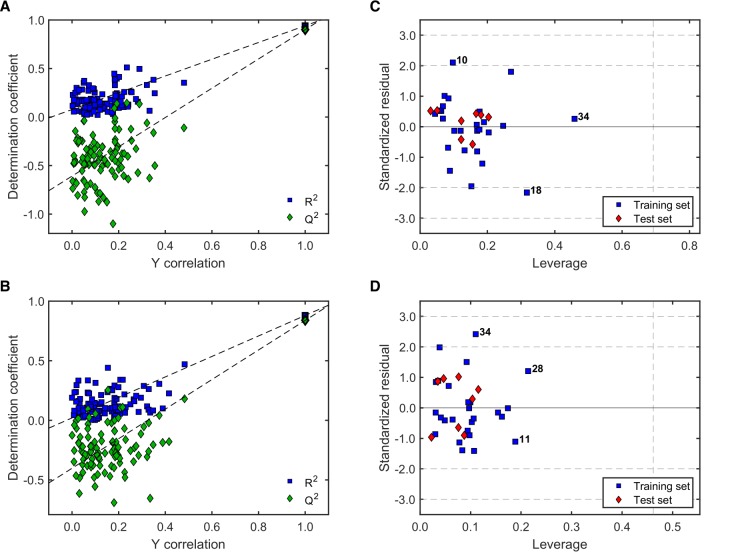
Evaluation of robustness and AD of the best models predicting antitrypanosomal activity. (**A**) Y-scrambling for M13-3; (**B**) Y-scrambling for M14-2; (**C**) Williams plot for M13-3; and (**D**) Williams plot for M14-2. In A and B, $R^{2}$ and $Q^{2}$ values for the model appear in the upper right corner with thicker marker edge. Horizontal dashed lines in C and D indicate 2σ and 3σ. The vertical dashed line represents *h** [50,51,52].

**Table 1 molecules-24-04358-t001:** Frequency of appearance of the main molecular descriptors affecting the selected quantitative structure-activity relationships (QSAR) models predicting antileishmanial activity.

Descriptor *	Frequency Per Total of Descriptors (%)	Frequency Per Number of Equations (%)
ASA^–^	12.9	54
PEOE_VSA_FPOL	11.4	48
chi1v	7.1	30
PEOE_VSA_FHYD	7.1	30
vsurf_D1	5.7	24

* See text for meaning. A full list of descriptors with their corresponding explanations is given in Appendix A.

**Table 2 molecules-24-04358-t002:** Definition of scoring functions to account for the overall quality of the QSAR models.

Function	Definition	Observations ^a^		
		*P*1	*P*2	*P*3 ^b^
***F*1**	$F1=\overset{7}{\underset{i=1}{\sum }}P1_{i}+\overset{3}{\underset{i=1}{\sum }}P2_{i}+\overset{2}{\underset{i=1}{\sum }}P3_{i}$	1) $R^{2}$ > 0.8 2) $Q^{2}$ > 0.5 3) $Q_{F1}^{2}$ > 0.6 4) $Q_{F2}^{2}$ > 0.5 5) $Q_{F3}^{2}$ > 0.5 6) CCC>0.85 7) $\bar{R_{m}^{2}}>0.5$	1) $\frac{R^{2}-R_{0}^{2}}{R^{2}}\;or\;\frac{R^{2}-R_{0}^{{}^{\prime }2}}{R^{2}}<0.1$ 2) $\left|R_{0}^{2}-R_{0}^{{}^{\prime }2}\right|<0.3$ 3) $\Delta R_{m}^{2}<0.2$	1) $MAE\leq 0.34\;and\;MAE+3\sigma_{MAE}\leq 0.68$ 2) $MAE\leq 0.51\;and\;MAE+3\sigma_{MAE}\leq 0.85$
*F*2	$F2=\overset{8}{\underset{i=1}{\sum }}P1_{i}+\overset{3}{\underset{i=1}{\sum }}P2_{i}+\overset{2}{\underset{i=1}{\sum }}P3_{i}$	1) $R^{2}-0.8$ 2) $Q^{2}-0.5$ 3) $Q_{F1}^{2}-0.6$ 4) $Q_{F2}^{2}-0.5$ 5) $Q_{F3}^{2}-0.5$ 6) CCC−0.85 7) $\bar{R_{m}^{2}\;}-0.5$ 8) $R^{2}-Q_{F2}^{2}\;$ $if\;R^{2}>Q_{F2}^{2}$ $or\;-Q_{F2}^{2}\;$ $if\;R^{2}<Q_{F2}^{2}$	1) $if\;\frac{R^{2}-R_{0}^{2}}{R^{2}}<\frac{R^{2}-R_{0}^{{}^{\prime }2}}{R^{2}}:$ $0.1-\frac{R^{2}-R_{0}^{2}}{R^{2}}$ $otherwise:\;0.1-\frac{R^{2}-R_{0}^{{}^{\prime }2}}{R^{2}}$ 2) $0.3-\left|R_{0}^{2}-R_{0}^{{}^{\prime }2}\right|$ 3) $0.2-\Delta R_{m}^{2}$	1) If MAE−0.51>0 or $MAE+3\sigma_{MAE}-0.85>0:$ MAE−0.51 $if\;MAE-0.51>MAE+3\sigma_{MAE}-0.85$ and $3\sigma_{MAE}-0.85$ $if\;MAE-0.51<MAE+3\sigma_{MAE}-0.85$ 2) If MAE−0.34>0 and $MAE+3\sigma_{MAE}-0.68>0$ Or if MAE−0.34<0 and $MAE+3\sigma_{MAE}-0.68<0:$ $\left(0.34-MAE\right)+\left(0.68-\left[MAE+3\sigma_{MAE}\right]\right)$
*F*3 ^c^	$F3=\frac{F2}{nd}\;if\;MAE<0.34\;$ $and\;MAE+3\sigma_{MAE}<0.68$	--	--	--

^a^ All the definitions of the involved parameters are found in the supporting information. ^b^ Thresholds as originally reported [40] (e.g., 0.1 × training set activity range = 0.34; 0.25× training set activity range = 0.85). ^c^ nd denotes the number of descriptors used in the model.

**Table 3 molecules-24-04358-t003:** Score values of QSAR models for antileishmanial activity of cinnamate ester analogues. The scores are defined in Table 2.

Model	TR ^a^	*Nd* ^b^	*F*1	*F*2 ^c^	*F*3 ^c^	Model	TR ^a^	*Nd* ^b^	*F*1	*F*2 ^c^	*F*3 ^c^
M1-1	1	3	8	1.49	-	M6-1	2	5	10	2.47	-
M1-2	1	3	6	−0.07	-	M6-2	2	5	12	**2.99**	**0.598**
M1-3	1	3	9	2.26	-	M6-3	2	5	10	2.48	-
M1-4	1	3	8	1.23	-	M6-4	2	5	12	**2.62**	0.525
M1-5	1	3	10	2.31	-	M6-5	2	5	12	**2.81**	**0.562**
M2-1	1	4	12	2.53	**0.632**	M7-1	3	3	9	1.64	-
M2-2	1	4	4	−1.00	-	M7-2	3	3	8	0.61	-
M2-3	1	4	11	2.08	-	M7-3	3	3	9	0.72	-
M2-4	1	4	12	2.18	**0.546**	M7-4	3	3	9	0.66	-
M2-5	1	4	9	1.82	-	M7-5	3	3	8	0.65	-
M3-1	1	5	6	−0.04	-	M8-1	3	4	9	1.45	-
M3-2	1	5	12	2.04	0.409	M8-2	3	4	7	−0.20	-
M3-3	1	5	10	2.20	-	M8-3	3	4	8	0.98	-
M3-4	1	5	7	1.54	-	M8-4	3	4	8	0.91	-
M3-5	1	5	3	−3.18	-	M8-5	3	4	7	0.63	-
M4-1	2	3	2	−6.56	-	M9-1	3	5	10	1.85	-
M4-2	2	3	2	−6.56	-	M9-2	3	5	10	2.34	-
M4-3	2	3	9	1.56	-	M9-3	3	5	9	1.64	-
M4-4	2	3	9	1.56	-	M9-4	3	5	9	1.67	-
M4-5	2	3	10	2.19	-	M9-5	3	5	9	1.67	-
M5-1	2	4	12	**2.77**	**0.692**	M10-1	3	6	11	**2.62**	-
M5-2	2	4	11	2.55	-	M10-2	3	6	11	2.60	-
M5-3	2	4	11	2.55	-	M10-3	3	6	9	1.87	-
M5-4	2	4	10	2.28	-	M10-4	3	6	9	1.88	-
M5-5	2	4	11	2.47	-	M10-5	3	6	10	2.02	-

^a^ TR = training set group: One and two for randomly selected training/test sets; and three for training set selected by Kennard-Stone algorithm. ^b^
*nd* = number of descriptors. ^c^ bold numbers highlight the top five models.

**Table 4 molecules-24-04358-t004:** QSAR equations of the best models for antileishmanial activity of the cinnamate ester analogues.

Model	Equation
M5-1	0.59011 − 0.0263602× ASA^–^ + 16.9485 × PEOE_VSA_FPOL + 0.162194 × a_IC − 0.00750344× vsurf_W3
M6-2	18.0466 − 0.025593× ASA^–^ − 17.8311× PEOE_VSA_FHYD − 0.00420797× Q_VSA_POL + 0.177629× a_IC − 0.00880367× vsurf_W3

**Table 5 molecules-24-04358-t005:** Frequency of appearance of the main molecular descriptors affecting the selected QSAR models predicting antitrypanosomal activity.

Descriptor *	Frequency Per Total of Descriptors (%)	Frequency Per Number of Equations (%)
vsurf_EWmin3	13.3	53.3
std_dim2	8.9	35.6
vsurf_EWmin2	4.4	17.8
ASA_P	2.8	11.1
chi1v_C	2.8	11.1
FCASA^+^	2.8	11.1
h_pKb	2.8	11.1
Q_VSA_FPPOS	2.8	11.1

* See text for meaning. A full list of descriptors with their corresponding explanations is given in Appendix A.

**Table 6 molecules-24-04358-t006:** Scoring values of QSAR models for antitrypanosomal activity of cinnamate ester analogues.

Model	TR ^a^	*Nd* ^b^	*F*1	*F*2 ^c^	*F*3 ^c^	Model	TR ^a^	*nd* ^b^	*F*1	*F*2 ^c^	*F*3 ^c^
M11-1	1	3	2	−32.2	-	M15-4	2	4	11	1.91	-
M11-2	1	3	2	−37.1	-	M15-5	2	4	11	1.99	-
M11-3	1	3	2	−31.6	-	M16-1	2	5	12	**2.62**	0.525
M11-4	1	3	2	−31.0	-	M16-2	2	5	12	**2.66**	0.531
M11-5	1	3	2	−33.2	-	M16-3	2	5	12	2.57	0.514
M12-1	1	4	12	2.33	0.584	M16-4	2	5	11	2.11	-
M12-2	1	4	12	2.17	0.543	M16-5	2	5	12	2.25	0.450
M12-3	1	4	12	2.42	**0.606**	M17-1	3	3	11	1.52	-
M12-4	1	4	12	**2.60**	**0.651**	M17-2	3	3	12	1.85	**0.616**
M12-5	1	4	12	**2.60**	**0.651**	M17-3	3	3	10	1.36	-
M13-1	1	5	12	2.39	0.479	M17-4	3	3	11	1.46	-
M13-2	1	5	12	2.59	0.518	M17-5	3	3	11	1.62	-
M13-3	1	5	12	**2.79**	0.558	M18-1	3	4	11	2.59	-
M13-4	1	5	12	2.43	0.486	M18-2	3	4	12	2.34	0.586
M13-5	1	5	12	2.60	0.519	M18-3	3	4	12	2.14	0.534
M14-1	2	3	11	2.56	-	M18-4	3	4	12	2.40	0.599
M14-2	2	3	12	2.32	**0.774**	M18-5	3	4	12	2.32	0.579
M14-3	2	3	11	1.48	-	M19-1	3	5	12	2.47	0.495
M14-4	2	3	11	2.18	-	M19-2	3	5	11	1.65	-
M14-5	2	3	11	1.93	-	M19-3	3	5	11	1.76	-
M15-1	2	4	11	2.31	-	M19-4	3	5	12	1.97	0.395
M15-2	2	4	11	1.46	-	M19-5	3	5	11	1.45	-
M15-3	2	4	7	0.65	-						

^a^ TR = training set group: One and two for randomly selected training/test sets; and three for training set selected by Kennard-Stone algorithm. ^b^
*nd* = number of descriptors. **^c^** bold numbers highlight the top five models.

**Table 7 molecules-24-04358-t007:** QSAR equations of the best models for antitrypanosomal activity of the cinnamate ester analogues.

Model	Equation
M13-3	6.76902 − 0.008234 × ASA^–^ − 15.0033 × Q_RPC^–^ + 3.09437× Q_VSA_FPPOS − 1.78347× std_dim2 − 0.801448× vsurf_EWmin3
M14-2	–1.45307 + 0.39037× chi1_C + 0.149959× lip_don − 0.771908× vsurf_EWmin3

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
