# Peer review of "A Comprehensive QSAR Study on Antileishmanial and Antitrypanosomal Cinnamate Ester Analogues"

_molecules, 2019, doi:10.3390/molecules24234358_

Round 1
Reviewer 1 Report
Comments
This manuscript described detailed QSAR studies on antileishmanial and antitrypanosomal cinnamate ester analogues using molecular fingerprints-based QSAR and theoretical descriptors-based QSAR methodologies. The whole manuscript was written well. I suggest this manuscript can be accepted after minor revision.
Several minor problems or suggestions are listed as follows:
1) I suggest the authors should add some recent QSAR studies in drug discovery or ecotoxicology that consider the comprehensive statistical criteria, such as “Hao et al., Ecotoxicology and Environmental Safety, 2019, 186, 109822”, “Sun et al., Molecules, 2018, 23, 2892” and “Fan et al., Int. J. Mol. Sci., 2018, 19, 3015”
2) The number of descriptors in proposed QSAR models should be less than the 1/5 of training set compounds. At least, the authors should mention and cite reference: “OECD, Guidance document on the validation of (Quantitative) structure–activity relationships [(Q)SAR] models, Organisation for Economic Co-Operation and Development, Paris, France, 2007”. The rule is also mentioned in three references listed above.
3) Besides, the critical hat value should be calculated as “h*= 3(w+1)/N” rather than “h*= 3w/N” in your manuscript. Also, the authors can read the recommended references listed above.
4) In the legend of Figure 1, the reference [13] should be deleted.
5) Why Q2F1 not included in your manuscript as a statistical parameter? Reference: Gramatica, P. et al. J. Chem. Inf. Model., 56, 1127-1131; Chirico et al., J. Chem. Inf. Model. 2011, 51, 2320-2335.
Author Response
Reviewer 1
This manuscript described detailed QSAR studies on antileishmanial and antitrypanosomal cinnamate ester analogues using molecular fingerprints-based QSAR and theoretical descriptors-based QSAR methodologies. The whole manuscript was written well. I suggest this manuscript can be accepted after minor revision.
Several minor problems or suggestions are listed as follows:
1) I suggest the authors should add some recent QSAR studies in drug discovery or ecotoxicology that consider the comprehensive statistical criteria, such as “Hao et al., Ecotoxicology and Environmental Safety, 2019, 186, 109822”, “Sun et al., Molecules, 2018, 23, 2892” and “Fan et al., Int. J. Mol. Sci., 2018, 19, 3015”
We thank the reviewer for this hint. We have included the more recent publication as a new reference in the revised version. It is now ref [48].
2) The number of descriptors in proposed QSAR models should be less than the 1/5 of training set compounds. At least, the authors should mention and cite reference: “OECD, Guidance document on the validation of (Quantitative) structure–activity relationships [(Q)SAR] models, Organisation for Economic Co-Operation and Development, Paris, France, 2007”. The rule is also mentioned in three references listed above.
We have also included a short statement that our work is in accordance with this rule of thumb and cited the appropriate reference. It is now ref [35]
3) Besides, the critical hat value should be calculated as “h*= 3(w+1)/N” rather than “h*= 3w/N” in your manuscript. Also, the authors can read the recommended references listed above.
We thank the reviewer for her/his scrutiny in detecting this error. The values have been recalculated and the diagrams modified accordingly. However, the overall result has not been affected.
4) In the legend of Figure 1, the reference [13] should be deleted.
It is not clear to us why the reviewer asks this. The reference is pointing to the publication in which the compounds‘ synthesis and analytical characterization are to be found. We have made this more clear in the figure legend; the reference is now [25].
5) Why Q2F1 not included in your manuscript as a statistical parameter? Reference: Gramatica, P. et al. J. Chem. Inf. Model., 56, 1127-1131; Chirico et al., J. Chem. Inf. Model. 2011, 51, 2320-2335.
This was a terminological mistake. In fact, Q2F1was already included in our work but termed R2pred. This is because some authors use divergent terms for these parameters. We have corrected this and now use the suggested terminology which is more consistent.
We wish to thank all reviewers for their time and efforts which helps us to improve this manuscript!
Reviewer 2 Report
This is a well written paper. I recommend this article for publication.
Author Response
Reviewer 2
This is a well written paper. I recommend this article for publication.
We thank this reviewer for the very positive assessment.
We wish to thank all reviewers for their time and efforts which helps us to improve this manuscript!
Reviewer 3 Report
This manuscript describes a systematic study on the antiparasitic activity (antileishmanial and antitrypanosomal) of the analogues of cinnamate esters. It is a continuation of the studies performed by the same group [Bernal et al, ChemMedChem, 2019, in press]. In the current manuscript a Quantitative Structure – Activity Relationships analyses were performed on the set of the recently published synthetic compounds. To define best model a set of scoring functions (newly introduced) accounting for fourteen validation parameters and different validation criteria was used. This approach is aimed at rational optimization of the structures of the compounds to be developed as candidates for the antiprotozoal pharmaceuticals for treatment of these Neglected Tropical Diseases.
General Comments
Data obtained through the PCA analyses are not fully clear for the reader – without any supplemental information the rationale for the following statement is not clear: ‘However, the compounds in red and yellow were consistently discriminated from each other by the third principal component (data not shown) which appeared to be related to the nature of the ester side chain.’ Moreover, why esters of methanol and branched alcohols were groupped together – it should be briefly explained in this section of the Results. Additionally, the color code applied has to be included in the Fig. 2 legend.
The biological sense of the parameter S has to be briefly explained in the following sentence: ‘Analysis of the corresponding S-line plot (Figure 2D) revealed that the structural keys contributing most to the variance in antileishmanial activity, and at the same time highly correlating with it, were number 140, 144, 150, 147, and 132.’ How these numbers ‘are related to the number of oxygen atoms in the molecule’? Moreover, a comment on the methodological difference between both approaches, i.e. previously used SAR maps [Bernal et al, ChemMedChem, 2019] and the S-line employed here, should be explained in the manuscript.
The general sense and the biological context of the ‘genetic algorithm’ used for selection of the best descriptors of the models should be explained in the manuscript. Table 1 and Table 5 captions should include the reference note to the Table S5 (explanation of the individual descriptors).
The source of the data (Table S1) on the experimental antileishmanial (Fig.3) and antitrypanosomal (Fig.5) activities has to mentioned in the legends to the respective figures.
Three new publications focused on antiprotozoal properties of the cinnamic derivatives have been published very recently, i.e. [Rodrigues et al., Eur J Med Chem. 2019], [Santos et al., Rev Soc Bras Med Trop. 2018], [da Silva et al., Chem Biol Drug Des. 2019]. These data have to be commented in the manuscript; it would of special interest for the audience to see the comments of the Authors on the structure of the compounds described in these publications in the context of the predictions of the QSAR models introduced here.
Conclussions should include answers to the following questions:
(i) which of the analyzed compounds appeared most promissing in the light of the QSAR models?
(ii) are there any new hits, comparing to [Bernal et a., 2019], with potential antiparasitic properties suggested by this approach?
(iii) in which way data presented here increase the knowledge comparing to the previous publication [Bernal et al, ChemMedChem, 2019]?
(iv) the authors have said previously, that ‘Efforts to evaluate the potential of the most potent antitrypanosomal compounds in in vivo models lacked success’ – does the current manuscript help to improve this state of the art? In the current manuscript Authors say that their approach ‘it aided to the comprehensive analysis of series of QSAR models obtained by GA/MLR with extraordinary quality of predictions of the antileishmanial and antitrypanosomal activity of the studied compounds’. In the opinion of this reviewer the statement on the extraordinary quality of predictions of any feature requires experimental verification while such data are not included in the manuscript. These issues have to be addressed in the manuscript.
(v) do the results obtain it this study verify the conclusion of the previous paper that 4-hydroxy-3-nitrophenyl cinnamic acid esters were the most promising hits against Leishmania
(vi) any comment why antiplasmodial aspect of the properties of the analyzed compounds [Bernal et a., 2019] was expelled from the current QSAR analysis would be of help to understand the rationale of the current study.
Minor remarks
l.95 – explain the meaning of P (cLogP) and spell out TPSA and MOE
Author Response
Reviewer 3
This manuscript describes a systematic study on the antiparasitic activity (antileishmanial and antitrypanosomal) of the analogues of cinnamate esters. It is a continuation of the studies performed by the same group [Bernal et al, ChemMedChem, 2019, in press]. In the current manuscript a Quantitative Structure – Activity Relationships analyses were performed on the set of the recently published synthetic compounds. To define best model a set of scoring functions (newly introduced) accounting for fourteen validation parameters and different validation criteria was used. This approach is aimed at rational optimization of the structures of the compounds to be developed as candidates for the antiprotozoal pharmaceuticals for treatment of these Neglected Tropical Diseases.
General Comments
Data obtained through the PCA analyses are not fully clear for the reader – without any supplemental information the rationale for the following statement is not clear: ‘However, the compounds in red and yellow were consistently discriminated from each other by the third principal component (data not shown) which appeared to be related to the nature of the ester side chain.’ Moreover, why esters of methanol and branched alcohols were groupped together – it should be briefly explained in this section of the Results. Additionally, the color code applied has to be included in the Fig. 2 legend.
A scores plot of PC3 vs PC2 has been added as Fig S1 to the supplemenary file so that the separation between red and yellow compound markers becomes obvious.
The second part of the question is difficult to anwer. We could only speculate that this has to do with the more spherical shape of methyl, isopropyl and tert-butyl groups which distinguishes them from the more elongated n-alkyl esters. However, we feel that this is would be too speculative and would rather not over-interpret the data. We have therefore not mentioned this speculation in the manuscript.
The biological sense of the parameter S has to be briefly explained in the following sentence: ‘Analysis of the corresponding S-line plot (Figure 2D) revealed that the structural keys contributing most to the variance in antileishmanial activity, and at the same time highly correlating with it, were number 140, 144, 150, 147, and 132.’ How these numbers ‘are related to the number of oxygen atoms in the molecule’? Moreover, a comment on the methodological difference between both approaches, i.e. previously used SAR maps [Bernal et al, ChemMedChem, 2019] and the S-line employed here, should be explained in the manuscript.
The S-line plot explains the impact of single variables on the overall model. We have made its meaning more clear in the text (lines 86-89) and cited reference [27] for further information.
SAR maps are purely non-quantitative representations which are by no means supported by statistics. This is different in the case of S-line plots. The magnitude of the p-Value (Y axis value) gives an estimate of the actual importance of a structural feature for activity. Thus, S-line plots give more reliable information in a quantitative manner than SAR maps can.
The general sense and the biological context of the ‘genetic algorithm’ used for selection of the best descriptors of the models should be explained in the manuscript. Table 1 and Table 5 captions should include the reference note to the Table S5 (explanation of the individual descriptors).
Genetic algorithms are a very widespread approach to variable selection in QSAR. This is known to any reader who is more deeply involved in QSAR and related approaches. We do not see the necessity to further explain its utilization and usefulness. Nevertheless, we have included further references [33-35] in case of further interest by the reader.
The source of the data (Table S1) on the experimental antileishmanial (Fig.3) and antitrypanosomal (Fig.5) activities has to mentioned in the legends to the respective figures.
This information has been added, as a service to the reader.
Three new publications focused on antiprotozoal properties of the cinnamic derivatives have been published very recently, i.e. [Rodrigues et al., Eur J Med Chem. 2019], [Santos et al., Rev Soc Bras Med Trop. 2018], [da Silva et al., Chem Biol Drug Des. 2019]. These data have to be commented in the manuscript; it would of special interest for the audience to see the comments of the Authors on the structure of the compounds described in these publications in the context of the predictions of the QSAR models introduced here.
We have added these references in the introduction, since they confirm that the cinnamic acid scaffold may be very useful to develop antiparasitic drugs. The compounds studied in these investigations, however, except being cinnamic acid related, are rather different from ours, i.e. they are outside applicability domain of our models so that no link can be constructed between these data and ours.
Moreover, different parasite species and life stages were studied by these authors. It is well known that especially Leishmania spp. (often even strains of the same species) and their different life forms show completely different susceptibility to drugs. Therefore, we will not speculate on any relationships of the mentioned studies with ours.
Conclussions should include answers to the following questions:
(i) which of the analyzed compounds appeared most promissing in the light of the QSAR models?
A QSAR study does not serve to prioritize compounds but to obtain a model that explains the structure-dependence of activity and to allow predictions for further compounds within the applicability domain. The most promising compounds we have so far are still the same as mentioned in our previous communication.
(ii) are there any new hits, comparing to [Bernal et a., 2019], with potential antiparasitic properties suggested by this approach?
Indeed, predictions for new compounds can be made on grounds of good QSAR models such as the ones presented here. This is, however, not the aim and scope of the presented work. The design of new compounds on the basis of these models (with hopefully even better activity) will be subject of subsequent communications.
(iii) in which way data presented here increase the knowledge comparing to the previous publication [Bernal et al, ChemMedChem, 2019]?
The data presented here add significant knowledge on quantitative relationships between molecular properties and activity which were not previously known. If fact this is the very essence of the meaning of „Q“ in QSAR. To identify such properties requires very thorough modelling and validation which is presented in this manuscript.
(iv) the authors have said previously, that ‘Efforts to evaluate the potential of the most potent antitrypanosomal compounds in in vivo models lacked success’ – does the current manuscript help to improve this state of the art? In the current manuscript Authors say that their approach ‘it aided to the comprehensive analysis of series of QSAR models obtained by GA/MLR with extraordinary quality of predictions of the antileishmanial and antitrypanosomal activity of the studied compounds’. In the opinion of this reviewer the statement on the extraordinary quality of predictions of any feature requires experimental verification while such data are not included in the manuscript. These issues have to be addressed in the manuscript.
Yes, this manuscript does improve the state-of-the-art since no quantitative relationships between structural and/or molecular features of such compounds and their antiprotozoal activity were previously known. We consider this a rather significant increase of knowledge about these interesting compounds.
Nevertheless, the statement cited by the reviewer has been slightly modified in order to make this more clear even to readers who are not so familiar with QSAR.
(v) do the results obtain it this study verify the conclusion of the previous paper that 4-hydroxy-3-nitrophenyl cinnamic acid esters were the most promising hits against Leishmania
A study such as the present one does not serve to confirm or verify the previous data or statements. These compounds were and still are the most promising among those compounds of this class against Leishmania donovani that presently exist. It is not the scope of a QSAR study which is BASED on data to verify those data.
(vi) any comment why antiplasmodial aspect of the properties of the analyzed compounds [Bernal et a., 2019] was expelled from the current QSAR analysis would be of help to understand the rationale of the current study.
Antiplasmodial activity of these compounds was generally very low except one compound that represented a special case (in terms of structure) and probably acts in a different way on the plasmodia. This compound was not included in the present study since it represents a structural outlier (i.e. is not within the applicability domain of the QSAR models). QSAR is based on statistics and it should be clear that it is not possible to create statistical models based on singular observations. Therefore, attempts to generate QSAR models for antiplasmodial activity within this set of compounds would not only be useless but scientific nonsense.
Minor remarks
l.95 – explain the meaning of P (cLogP) and spell out TPSA and MOE
Even though these terms are well known among scientists involved in QSAR and medicinal chemistry who will probably be the main readership of this article, we have spelled them out in the first instance of mentioning.
Reviewer 4 Report
The manuscritp is really well-conceived and discussed byt unfortunately qsar studies performed on 34 compounds (divided into atraining set and test set) is poorly significant. They should include more compounds, also featuring different chemo-types.
Author Response
Reviewer 4
The manuscritp is really well-conceived and discussed byt unfortunately qsar studies performed on 34 compounds (divided into atraining set and test set) is poorly significant. They should include more compounds, also featuring different chemo-types.
We disagree with this reviewer. It is a misconception that a QSAR study with 34 compounds must per se be poorly significant. In fact, the statistical significance of such models is proven by the thorough validation which, in our case, yielded very good results. The reason why so many researchers spend a lot of effort on model validation (which is very thoroughly done in our work) is that yes, models may be poorly significant, if not properly validated. In our case, all the statistics, including the applicability domain assessment, point towards good models. The reviewer should not overlook that there is considerable chemical diversity in this set of related compounds, i.e. a rather significant and coherent chemical space is covered by them. This is –in our view- more important than having more observations within a less coherent chemical space (this is probably what the reviewer means by „different chemo-types“).
We wish to thank all reviewers for their time and efforts which helps us to improve this manuscript!
Round 2
Reviewer 4 Report
I see what you mean but if you look your dataset, all the compounds share the same main scaffold and limited number of groups included as R substituents.
In any case, the statical analysis is well-performed and conceived, as I have already written in my previous comments.